# Variation in Diet Patterns of the Invasive Top Predator *Sander lucioperca* (Linnaeus, 1758) across Portuguese Basins

Diogo Ribeiro [1],*, Christos Gkenas [1], João Gago [1,2] and Filipe Ribeiro [1]

1   MARE, Centro de Ciências do Mar e do Ambiente, Faculdade de Ciências da Universidade de Lisboa, 1749-016 Lisboa, Portugal; cgkenas@fc.ul.pt (C.G.); joao.gago@esa.ipsantarem.pt (J.G.); fmvribeiro@gmail.com (F.R.)
2   Escola Superior Agrária—Instituto Politécnico de Santarém, Quinta do Galinheiro—S. Pedro, 2001-904 Santarém, Portugal
*   Correspondence: diogorrribeiro@gmail.com

**Abstract:** The introduction of non-native species is recognized as a major threat to biodiversity, particularly in freshwater ecosystems. Pikeperch *Sander lucioperca*, is a recent invader to Portugal, primarily providing commercial and angling interest. The aim of this work was to study the diet of this top predator across Portuguese basins and to evaluate its potential impact on recipient ecosystems. In total, 256 pikeperch stomachs from seven basins were examined, of which 88 ($n$ = 34%) were empty. Pikeperch diet was dominated by *R. rutilus*, *M. salmoides* and Diptera in northern populations, while *A. alburnus*, *P. clarkii* and Atyidae were important prey in more humid highlands. Variation in diet was most strongly linked to latitude and ontogeny, with both size classes showing signs of cannibalism. The population niche breadth remained low and was accompanied by higher individual diet specialization, particularly in northern populations. Pikeperch dietary patterns denoted an opportunistic ability to use locally abundant prey in each ecosystem, and was size dependent, with larger individuals becoming more piscivores, causing a higher impact in the lotic systems. This first perspective about the pikeperch diet presents a very broad view of the feeding traits of this non-native predator across Portugal, being very important to deepen our knowledge about the impact of these introduced piscivores.

**Keywords:** freshwaters; pikeperch; trophic ecology; diet specialization; non-native fish

## 1. Introduction

Freshwater ecosystems are highly imperiled worldwide due to multiple pressures that result in declines of aquatic biota [1,2]. Not surprisingly, freshwater taxa such as fish or molluscs present as being under threat at high rates. When compared with terrestrial or marine ecosystems, freshwater ecosystems are generally overlooked in biodiversity studies, although they contain higher biodiversity per area and face higher declines [3,4]. Biological invasions of non-native species within freshwater ecosystems are worrisome [1,5–8], and the introduction rate of new species has been increasing [9].

Freshwater fishes are the most introduced vertebrate group, due to aquaculture, angling and commercial fisheries and for ornamental trade purposes [10–14]. In European freshwaters, recreational fisheries are one of the main drivers of fish introduction, mostly focusing on top predatory fish, particularly in southern peninsulas [15,16]. In Portuguese freshwater ecosystems, the introduction of non-native fish (NNF) has reached one new species every two years [17,18]. From a total of 62 fish species currently existent in Portuguese watersheds, 19 are non-native and some are top predators, recently established, with potentially high deleterious impacts to fish communities and aquatic food webs [19,20]. In addition, most of the native fish communities in the Iberian Peninsula evolved without predatory fish, so the introduction of predatory fish could have strong effects on the fish

community [20]. The impacts of introduced predators can be devastating, leading to local native fish species extinctions [21,22] and modifying food webs and ecosystems [23,24].

Pikeperch *Sander lucioperca* (Linnaeus, 1758) is a predatory fish native to central Europe and western Asia, that has been intentionally introduced to fresh and brackish waters in Europe, Asia and Africa [25]. This species was introduced to the Iberian Peninsula in the 1970s on Catalonian reservoirs [26] and in 1998 was recorded in mainland Portugal [27]. It has now extended its distribution to most parts of the hydrographic network of Portuguese watersheds, where it has important angling and commercial interest [28–30]. Annually, pikeperch consumes several times its own bodyweight of prey, implying an impact on native aquatic communities [31]. Despite exhibiting ontogenetic shifts in its diet, it rapidly attains piscivory during its first year of life [32,33]. The predatory behavior of pikeperch has been well studied both within their native and non-native ranges (e.g., [34–38]). However, there is little information on the trophic ecology of pikeperch in relation to novel environments, which could depict distinct impacts and effects on native fish communities.

The present study uses a spatial approach to compare the diet of *S. lucioperca* across different populations in Portugal, subjected to different environmental settings. Specifically, we aim to (i) determine the diet composition of the species, (ii) quantify spatial and size-related changes in diet and (iii) identify differences in niche breadth at the population and individual level. This information will contribute to assess the potential impact of the species in an endemic rich area, such as the Iberian Peninsula, where there is an urgent need to evaluate the impact of non-native predatory fish.

## 2. Materials and Methods

### 2.1. Field Sampling and Laboratory Procedures

Pikeperch were sampled during 2017 and 2018, from April to October in selected river basins either in lotic or lentic habitats (Figure 1) and covering a total of 11 populations across mainland Portugal. The extensive coverage area of the fish population selection is important to evaluate the geographic dietary differences across the continental area of Portugal. Additionally, in these sites there was considerable commercial fishing pressure to pikeperch which provided easy access to fishes. Although this plan allowed us to use a higher geographic coverage, the sampling period was fishermen-dependent and so it could not be standardized. The majority of the specimens used in this study were provided by fishermen that used overnight gillnets of 80–150 mm mesh size as a fishing technique. Some juveniles were also captured by standardized electrofishing (300–500 V, 1–5 A).

In the laboratory, specimens were measured (Standard Length—SL, to nearest 1 mm), weighed (Eviscerated Weight—EW, to the nearest 0.01 g) and their stomachs were dissected, labelled and preserved immediately by freezing until stomach content analysis. Prey items were examined under a binocular dissecting microscope, identified to the lowest readily recognizable taxon (species and family was achieved for fishes and crayfish; order and family for insects) and counted. Identifications followed published keys and literature [39–45]. Additionally, for prey item identification, a reference collection was created with the bony parts of prey species (otoliths, scales, pharyngeal teeth). In total, we examined the stomach contents of 256 pikeperch, ranging from 9.4 to 60.3 cm SL, of which 88 individuals ($n = 34\%$) had empty stomachs which were discarded to avoid confounding effects in the analysis of diet structure [46].

To identify size-related diet shifts, fish were grouped into two standard length classes: I $\leq$ 25 cm; II > 25 cm and differences in the contribution of prey items to the diet of individuals in each size class were based on the stomach content analysis. The definition of this limit is related with the onset of reproduction, where all the individuals belonging to Class I (ages 0 to 3) are juvenile fish, while most of Class II are adult fish (fish older than 3 years, see [29]). Given the small sample size of some fish captures (e.g., Rio Ave), to provide sufficient power in analyses, only populations with more than 5 individuals were included. For analysis, prey items were grouped into thirteen categories: *Alburnus alburnus*, *Sander lucioperca*, *Rutilus rutilus*, *Micropterus salmoides*, *Lepomis gibbosus*, Mugilidae, Diptera, Ephemeroptera,

Odonata, Atyidae, *Procambarus clarkii*, "Other Fish" and "Other insects". These categories were defined following taxonomic affinities, so that each category contributed to >1% of the total prey in at least one site. The categories "Other Fish" included native cyprinids (nase *Pseudochondrostoma* spp.), non-native Cyprinid, Iberian gudgeon (*Gobio lozanoi*) and rarely found migratory species, such as Big-scale sand smelt (*Atherina boyeri*), European seabass (*Dicentrarchus labrax*), European flounder (*Platichthys flesus*), and "Other insects" comprised rare and unidentified prey.

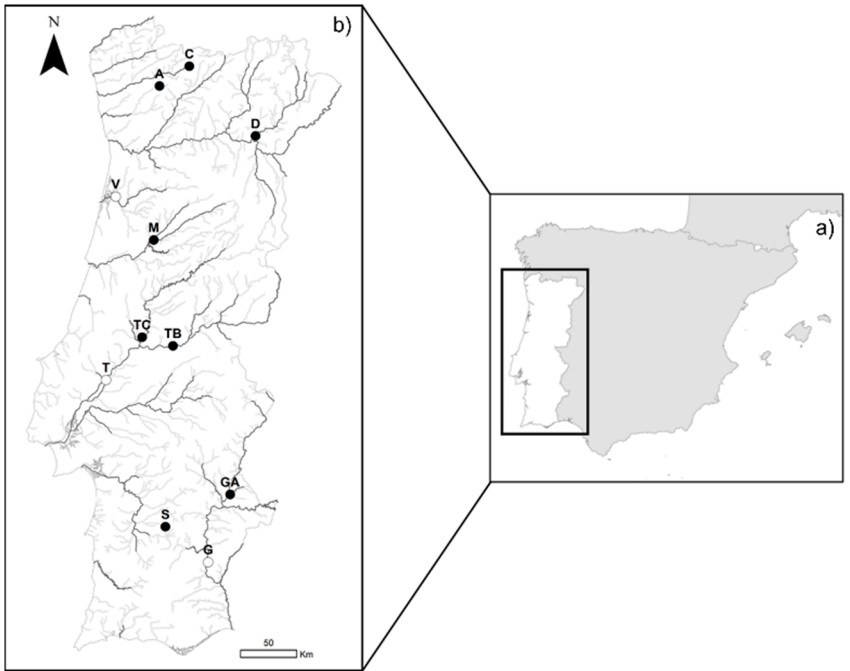

**Figure 1.** Map showing locations of pikeperch *Sander lucioperca* populations in (**a**) Iberian Peninsula and (**b**) mainland Portugal. White circles correspond to lotic populations and black ones to lentic populations. From north to south (Drainage—Location/Habitat): C, Cávado River—"Alto do Rabagão" reservoir; A, Ave River—"Ermal"—reservoir; D, Douro River—"Foz do Sabor" reservoir; V, Vouga River—lotic section near "Angeja"; M, Mondego River—"Aguieira" reservoir; TC, Tagus River—"Castelo de Bode" reservoir; TB, Tagus River—"Belver" reservoir; T, Tagus River—lotic section near "Santarém"; S, Sado River—"Penedrão" reservoir; GA, Guadiana River—"Alqueva" reservoir; G, Guadiana River—lotic section near "Mértola".

*2.2. Data Analyses*

We described diet composition at each site using two conventional indices, the frequency of occurrence (F%) which is the proportion of non-empty stomachs containing a particular prey category, and the numerical frequency (N%), which is the proportional count of each prey category relative to the total prey count among fish [47].

We used non-metric multidimensional scaling (NMDS) analyses to visualize patterns in diet composition in the whole diet composition of pikeperch among the sites using the Bray–Curtis similarity coefficient [48]. Prior to analysis, prey proportions were determined for each individual in relation to the total prey in its stomach, and were square root transformed to reduce the influence of abundant prey in the analysis. Because sample sizes varied among sites and size classes, we averaged the mean location (i.e., centroids) from the NMDS coordinates of all the individual fish to assess geographic changes in the diet composition. Ordination results were considered to be sufficiently described in two dimensions when stress was <0.2 [49]. Finally, we performed linear regression analysis to test for the variation in diet composition with latitude, using the centroids from the NMDS scores in the first two axes as response variables in the analysis.

We then used Permutational Multivariate Analysis of Variance (PERMANOVA) with 9999 permutations based on a dissimilarity Bray-Curtis matrix [50] to test for differences between the diets of pikeperch, using the sites and two size classes (I $\leq$ 25 cm, II > 25 cm SL) as factors. Next, the Indicator Value Index (IndVal) [51] was applied to obtain the prey item indicators for sites and sites-plus-size classes. The IndVal is based on a comparison of the specificity (the relative abundance of each food resource in each group or factor) and fidelity (the relative frequency of each food resource in each group or factor), that are being tested in different groups selected a priori [52]. The greater the specificity and fidelity of an item to a particular group, the greater the value of the indicator; and this method proves robust to differences within the group, sample sizes, and differences in the abundance between the groups [53].

Niche breadth was assessed at the population and individual level, using animal prey. Population breadth was determined according to [54], as follows:

$$B = \left( \sum p_j^2 \right)^{-1} \tag{1}$$

where $p_j$ is the proportion of prey category $j$ in the diet. The Levin's index ranges from 1 to $n$, where $n$ is the number of prey categories; it is at a minimum when there is only a prey category in the diet, and at a maximum when proportions are the same in all prey categories, indicating there is no discrimination among prey categories [55].

Individual specialization (IS) was determined using the proportional similarity index ($PS_i$), between each individual's prey proportions and the averaged population diet distribution, using the equation by [56], as follows:

$$PS_i = 1 - 0.5 \sum_{j=1} \left| p_{ij} - q_j \right| \tag{2}$$

where $p_{ij}$ is the proportion of prey category $j$ in the diet of individual $i$, and $q_j$ is the proportion of prey category $j$ in the population as a whole.

This index compares each individual's diet to that of the entire population, with values ranging between 0 and 1. For individuals specializing on single or few prey types, the $PS_i$ values tend to be low, whereas for individuals that consume resources in a similar proportion as the entire population, the $PS_i$ values approach 1 [56]. To evaluate variation in niche indices ($B$ and IS) between populations, we conducted 1000 bootstrap resamplings of the data for each case. Differences were significant when the 95% confidence intervals for estimates did not overlap. All analyses were conducted using the R software [57], and the significance of statistical testing was assessed at $p < 0.05$.

## 3. Results

In total, we analyzed 168 stomachs and 609 prey items, with sample sizes per site varying between 5 and 31 stomachs and 12–339 prey (Table 1). *Alburnus alburnus* was found in fish stomachs with the highest frequency (28%), followed by *S. lucioperca* (25.6%). Diptera larvae were the most abundant group of prey eaten by pikeperch with a contribution of 54.2%, followed by *A. alburnus* and *S. lucioperca,* but in small numbers (10.7% and 8.9%, respectively).

Diet composition varied considerably among sites and size classes (Table 1). Particularly in northern basins, large individuals consumed mostly *R. rutilus* (27.7; 50%) in the lentic sites Douro and Cávado and also preyed on high proportions of Diptera (84.6%) in the lentic Mondego and Mugilidae (42.9%) and in the lotic Vouga. Conversely, in small individuals, *M. salmoides* (100%) dominated the diet in the lentic Mondego, with Diptera making only 26.9% of the total prey in the lentic Douro, but dominated the diet in the lentic Cávado (66.7%) and the lotic Vouga (93.5%). Cannibalism occurred for both small and large individuals and increased with individual size in the lentic Douro (30.8%; 36.2%, respectively) and the lotic Vouga (0.9%; 14.3%, respectively).

**Table 1.** Variation in the numeric frequency (%) and in the frequency of occurrence (% in brackets) of prey categories consumed by class I (≤25 cm SL) and class II (>25 cm SL) pikeperch *Sander lucioperca* populations in Portuguese basins. Total fish contains the number of pikeperch stomachs analyzed (*n*) with empty stomachs presented in brackets, per population. Mean population Standard Length (SL), with size range, minimum and maximum values (min–max) presented for each analyzed population. Sites are ordered by decreasing latitude. Population acronyms according to Figure 1.

| Prey Categories | Overall | C Class I | C Class II | D Class I | D Class II | V Class I | V Class II | M Class I | M Class II | TC Class II | TB Class I | TB Class II | T Class I | T Class II | S Class I | S Class II | GA Class II | G Class II |
|---|---|---|---|---|---|---|---|---|---|---|---|---|---|---|---|---|---|---|
| *A. alburnus* | 10.7 (28.0) | 0.0 (0.0) | 0.0 (0.0) | 0.0 (0.0) | 21.3 (31.8) | 0.3 (5.3) | 0.0 (0.0) | 0.0 (0.0) | 0.0 (0.0) | 20.6 (25.0) | 0.0 (0.0) | 76.9 (91.7) | 0.0 (0.0) | 18.2 (33.3) | 0.0 (0.0) | 6.7 (8.3) | 52.4 (72.7) | 25.0 (40.0) |
| *S. lucioperca* | 8.9 (25.6) | 0.0 (0.0) | 0.0 (0.0) | 30.8 (66.7) | 36.2 (54.5) | 0.9 (15.8) | 14.3 (28.6) | 0.0 (0.0) | 0.0 (0.0) | 5.9 (12.5) | 0.0 (0.0) | 0.0 (0.0) | 5.0 (9.1) | 9.1 (16.7) | 100.0 (100.0) | 73.3 (75.0) | 0.0 (0.0) | 8.3 (20.0) |
| *R. rutilus* | 3.4 (6.5) | 33.3 (50.0) | 50.0 (55.6) | 0.0 (0.0) | 27.7 (27.3) | 0.0 (0.0) | 0.0 (0.0) | 0.0 (0.0) | 0.0 (0.0) | 0.0 (0.0) | 0.0 (0.0) | 0.0 (0.0) | 0.0 (0.0) | 0.0 (0.0) | 0.0 (0.0) | 0.0 (0.0) | 0.0 (0.0) | 0.0 (0.0) |
| Mugilidae | 1.1 (3.6) | 0.0 (0.0) | 0.0 (0.0) | 0.0 (0.0) | 0.0 (0.0) | 0.0 (0.0) | 42.9 (71.4) | 0.0 (0.0) | 0.0 (0.0) | 0.0 (0.0) | 0.0 (0.0) | 0.0 (0.0) | 0.0 (0.0) | 0.0 (0.0) | 0.0 (0.0) | 0.0 (0.0) | 0.0 (0.0) | 8.3 (20.0) |
| *M. salmoides* | 2.1 (5.4) | 0.0 (0.0) | 7.1 (11.1) | 0.0 (0.0) | 0.0 (0.0) | 0.0 (0.0) | 0.0 (0.0) | 100.0 (100.0) | 0.0 (0.0) | 20.6 (18.8) | 0.0 (0.0) | 0.0 (0.0) | 0.0 (0.0) | 0.0 (0.0) | 0.0 (0.0) | 0.0 (0.0) | 4.8 (9.1) | 0.0 (0.0) |
| *L. gibbosus* | 1.3 (4.8) | 0.0 (0.0) | 21.4 (22.2) | 0.0 (0.0) | 0.0 (0.0) | 0.0 (0.0) | 0.0 (0.0) | 0.0 (0.0) | 0.0 (0.0) | 11.8 (25.0) | 0.0 (0.0) | 0.0 (0.0) | 0.0 (0.0) | 0.0 (0.0) | 0.0 (0.0) | 0.0 (0.0) | 0.0 (0.0) | 8.3 (20.0) |
| O. Fish | 2.6 (9.5) | 0.0 (0.0) | 0.0 (0.0) | 0.0 (0.0) | 2.1 (4.5) | 0.6 (10.5) | 7.1 (14.3) | 0.0 (0.0) | 0.0 (0.0) | 2.9 (6.3) | 0.0 (0.0) | 2.6 (4.2) | 30.0 (54.5) | 18.2 (33.3) | 0.0 (0.0) | 6.7 (8.3) | 0.0 (0.0) | 8.3 (20.0) |
| Diptera | 54.2 (10.7) | 66.7 (100.0) | 21.4 (33.3) | 26.9 (33.3) | 0.0 (0.0) | 93.5 (52.6) | 0.0 (0.0) | 0.0 (0.0) | 84.6 (33.3) | 0.0 (0.0) | 0.0 (0.0) | 0.0 (0.0) | 0.0 (0.0) | 0.0 (0.0) | 0.0 (0.0) | 0.0 (0.0) | 14.3 (9.1) | 0.0 (0.0) |
| Ephemeroptera | 1.8 (3.0) | 0.0 (0.0) | 0.0 (0.0) | 19.2 (22.2) | 10.6 (9.1) | 0.0 (0.0) | 0.0 (0.0) | 0.0 (0.0) | 0.0 (0.0) | 2.9 (6.3) | 0.0 (0.0) | 0.0 (0.0) | 0.0 (0.0) | 0.0 (0.0) | 0.0 (0.0) | 0.0 (0.0) | 0.0 (0.0) | 0.0 (0.0) |
| Odonata | 1.3 (2.4) | 0.0 (0.0) | 0.0 (0.0) | 0.0 (0.0) | 0.0 (0.0) | 2.5 (21.1) | 0.0 (0.0) | 0.0 (0.0) | 0.0 (0.0) | 0.0 (0.0) | 0.0 (0.0) | 0.0 (0.0) | 0.0 (0.0) | 0.0 (0.0) | 0.0 (0.0) | 0.0 (0.0) | 0.0 (0.0) | 0.0 (0.0) |
| Atyidae | 7.4 (12.5) | 0.0 (0.0) | 0.0 (0.0) | 19.2 (22.2) | 2.1 (4.5) | 0.3 (5.3) | 35.7 (14.3) | 0.0 (0.0) | 0.0 (0.0) | 0.0 (0.0) | 100.0 (100.0) | 15.4 (25.0) | 65.0 (54.5) | 54.5 (16.7) | 0.0 (0.0) | 0.0 (0.0) | 0.0 (0.0) | 41.7 (20.0) |
| *P. clarkii* | 3.1 (5.4) | 0.0 (0.0) | 0.0 (0.0) | 0.0 (0.0) | 0.0 (0.0) | 0.0 (0.0) | 0.0 (0.0) | 0.0 (0.0) | 15.4 (66.7) | 29.4 (31.3) | 0.0 (0.0) | 0.0 (0.0) | 0.0 (0.0) | 0.0 (0.0) | 0.0 (0.0) | 6.7 (8.3) | 28.6 (9.1) | 0.0 (0.0) |
| O. insects | 2.0 (6.0) | 0.0 (0.0) | 0.0 (0.0) | 3.8 (11.1) | 0.0 (0.0) | 1.8 (21.1) | 0.0 (0.0) | 0.0 (0.0) | 0.0 (0.0) | 5.9 (12.5) | 0.0 (0.0) | 5.1 (8.3) | 0.0 (0.0) | 0.0 (0.0) | 0.0 (0.0) | 6.7 (8.3) | 0.0 (0.0) | 0.0 (0.0) |
| Total prey | 609 | 3 | 14 | 26 | 47 | 325 | 14 | 4 | 3 | 34 | 3 | 39 | 20 | 11 | 8 | 15 | 21 | 12 |
| Total fish (*n*) | 168 (88) | 2 (3) | 9 (16) | 9 (7) | 22 (3) | 19 (9) | 7 (1) | 4 (1) | 3 (5) | 16 (7) | 2 (0) | 24 (8) | 11 (3) | 6 (4) | 6 (3) | 12 (8) | 11 (9) | 5 (1) |
| SL cm | 29.5 | 23.6 | 34.6 | 19.8 | 32.0 | 12.2 | 38.0 | 21.5 | 42.7 | 40.5 | 19.6 | 29.6 | 15.8 | 29.2 | 22.2 | 39.3 | 40.2 | 41.3 |
| min–max | 9.4–60.3 | 22.9–24.2 | 25.4–53.7 | 12.4–24.1 | 28.1–39.2 | 9.4–19.0 | 31.8–60.3 | 20.2–22.9 | 31.6–48.5 | 25.7–52.2 | 15.9–23.2 | 25.1–38.2 | 9.5–24.2 | 27.4–30.8 | 19.6–23.9 | 25.2–56.6 | 32.4–43.2 | 31.0–51.0 |

In southern populations, *A. alburnus* dominated the diet of large pikeperch in the reservoirs of Alqueva (52.4%) and Belver (76.9%), with contributions of high proportions of Atyidae in the diet of the lotic sites Guadiana (41.7%) and Tagus (54.5%). Additional important prey was *P. clarkii* in the reservoirs of Castelo de Bode (29.4%) and Alqueva (28.6%). Small pikeperch preyed mostly on Atyidae in the reservoir Belver (100%) and the lotic Tagus (65%), also including "other fish" (30%) as important prey. Cannibalism was prevalent in the lentic Sado for both small (100%) and large (73.3%) pikeperch, but was found in lower numbers in the populations of the reservoir Castelo de Bode and the lotic sites Guadiana and Tagus (Table 1).

The NMDS biplot highlighted considerable diet variation in prey use among sites, with a slight separation of the lotic sites (Vouga, Tagus and Guadiana) and depicting a latitudinal gradient (Figure 2). Variation in NMDS1 scores showed no positive association with latitude ($F_{1,8}$ = 0.51, $R^2$ = −0.06, $p$ = 0.495), but there was a positive geographic pattern (e.g., north–south trend) between the latitude and diet composition for NMDS2 ($F_{1,8}$ = 8.19, $R^2$ = 0.44, $p$ = 0.021).

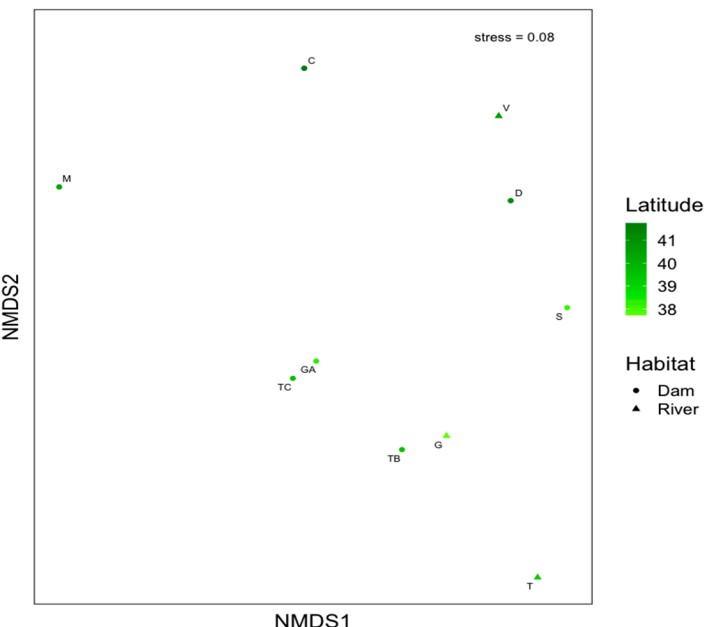

**Figure 2.** Nonmetric multidimensional scaling (NMDS) ordination of the diet of pikeperch *Sander lucioperca* populations in Portuguese basins using averaged mean locations (i.e., centroids). Symbols for sites are shaded by latitude from north (dark) to south (lighter).

The results of the two-way PERMANOVA (Table 2) showed that there was a significant interaction between sites x size classes (Pseudo $F_{6,151}$ = 2.38, $p$ < 0.001) and also differences in the diet associated with sites (Pseudo $F_{9,151}$ = 6.56, $p$ < 0.001) and size classes (Pseudo $F_{1,151}$ = 3.14, $p$ = 0.004).

**Table 2.** Permutational Multivariate Analysis of Variance results in the diet composition of pikeperch *Sander lucioperca* populations in Portuguese basins.

| Parameter | df | Pseudo *F*-Ratio | *p*-Value |
|---|---|---|---|
| Sites * Size | 6 | 2.38 | <0.0001 |
| Sites | 9 | 6.56 | <0.0001 |
| Size | 1 | 3.14 | 0.004 |
| Residual | 151 | | |

The most significant food items that contributed to the sites-plus-size class differentiation, were indicated by IndVal (Table 3). In northern populations, differences in prey

use were consistently related to contributions of Mugilidae and *P. clarkii* to large individuals from the Vouga and Mondego, but Diptera and *M. salmoides* were also important contributors for small pikeperch in the Cávado and Mondego. In southern populations, differentiation resulted from the contribution of *A. alburnus* to large individuals in the Belver reservoir, with other contributors being important for small pikeperches including Atyidae in the Belver reservoir, and "Other fish" and *S. lucioperca* in the river Tagus and Sado reservoirs, respectively.

**Table 3.** Indicator Value (IndVal), *p*-values and Frequency of prey items consumed by pikeperch *Sander lucioperca* populations in Portuguese basins between sites and size classes (I ≤ 25 cm, II > 25 cm SL). Population acronyms according to Figure 1.

| Prey Items | Site/Size Class | IndVal | *p*-Value | Frequency |
|---|---|---|---|---|
| Diptera | C (I) | 0.381 | 0.016 | 18 |
| Mugilidae | V (II) | 0.545 | 0.002 | 6 |
| *M. salmoides* | M (I) | 0.754 | 0.001 | 9 |
| *P. clarkii* | M (II) | 0.432 | 0.009 | 9 |
| Atyidae | TB (I) | 0.466 | 0.002 | 21 |
| *A. alburnus* | TB (II) | 0.266 | 0.001 | 47 |
| O. fish | T (I) | 0.195 | 0.054 | 16 |
| *S. lucioperca* | S (I) | 0.309 | 0.001 | 43 |

The contribution of non-native prey in the diet of pikeperch populations was very high, ranging from 53% in the lotic Tagus to 100% in the lentic sites Cávado, Mondego, Sado and Guadiana (Figure 3). Contrastingly, native fish prey was found only in the lotic sites Vouga (47%), Tagus (42%) and Guadiana (29%) and accounted with lower percentages in lentic sites.

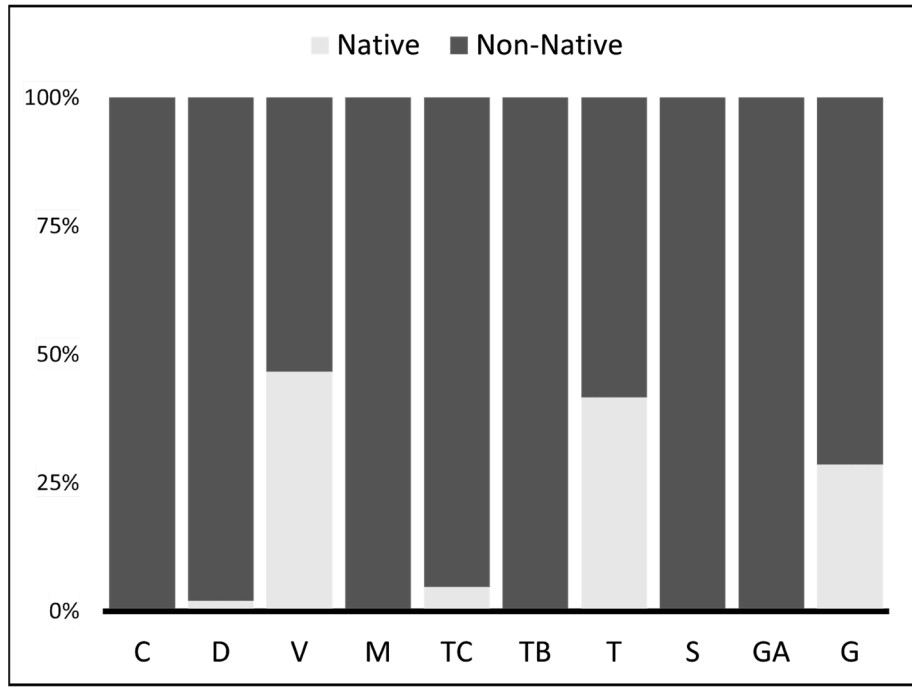

**Figure 3.** Consumption on Native vs. Non-Native fish preys by pikeperch *Sander lucioperca* populations in Portuguese basins. Light grey (Native), dark grey (Non-Native).

There was no substantial variation in the dietary niche breadth among populations of pikeperch (Figure 4). Levin's index showed a similar and narrow pattern, with average values ranging from one (lentic populations in the Mondego and Guadiana) to 1.32 (Castelo de Bode reservoir). The degree of individual specialization in prey use showed some

considerable variation among populations. Pikeperch individuals tended to be more specialized in the northern than in southern populations (Figure 4). Nevertheless, pikeperch showed higher specialization values in the Castelo de Bode reservoir and the lotic Guadiana than in the lentic Sado and the Belver reservoir.

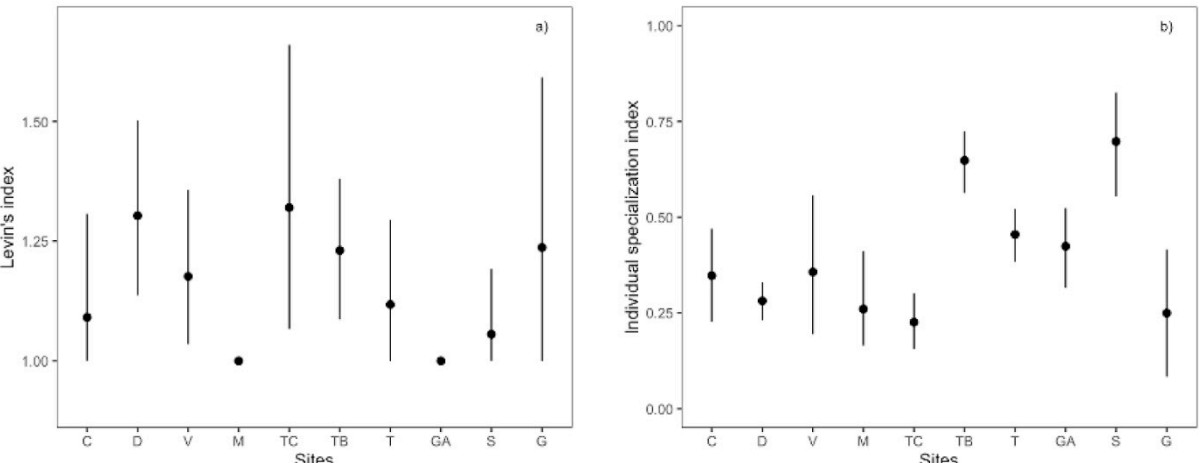

**Figure 4.** Population and individual niche breadth of the pikeperch *Sander lucioperca* populations in Portuguese basins: (**a**) Levin's index) and (**b**) individual specialization index. Error bars represent bootstrapped 95% confidence intervals for each index. Sites are ordered by decreasing latitude. Population acronyms according to Figure 1.

## 4. Discussion

Our results highlighted a variation in dietary patterns of *Sander lucioperca*, displaying a narrow niche breadth at both population and individual levels, and a highly opportunistic predatory behavior that is likely to be associated with the prey availability at a local level, which was not evaluated in this study. We observed a latitudinal gradient pattern on prey use by pikeperch, suggesting flexibility in its utilization of available prey resources. Ontogenetic fluctuations in prey use were also found, with small individuals feeding predominantly on insects, small macroinvertebrates and fish, while large specimens foraged mainly on fish and crayfish and with both size classes showing signs of cannibalism in six populations. These results are coincident with those found in lakes [58,59] and reservoirs [36,60], suggesting that fish prey constitute a staple prey for pikeperch across its non-native range. Furthermore, the majority of the pikeperch diet was composed of non-native species, although the presence of native fish preys (e.g., Mugilidae, Big-scale sand smelt, Unidentified nase, European seabass, European flounder) was found but in low numbers, but in lotic populations could reach half of the preys. Taken together, these findings suggest the importance of spatial comparisons in diet studies, supporting that non-native piscivorous fish may cause different impacts in relation to local prey richness and abundance.

The current work evaluates pikeperch dietary patterns from different populations that were collected across a wide spatial gradient, using different methodologies, encompassing distinct months and involving a limited number of individuals in some populations. However, the results presented here are generally consistent with other studies conducted in Spain and France, supporting the current observed patterns adjusted to Iberian watersheds [36,60]. The sampling period of this study might have influenced the results; however, most of the studied populations were lentic ones, related to reservoirs that are more stable environments with a limited community seasonal change [36]. Specifically, it would be important to widen the study period to encompass the entire year in riverine populations, to evaluate the predation effect on anadromous fish that spawn during the winter months [20]. Although limited sample size in some populations may have induced some shortcomings, it is unlikely to have any significant effects given the high individual specialization observed. The lack of a detailed key to identify fish bones hampered species

identification in a few cases (Table 1), where identification was only possible due to the creation of a fish bone collection with eleven species associated with previous fish keys (e.g., [44]). However, we recognize that molecular techniques (e.g., DNA metabarcoding) could improve our prey resolution and unveil a wider prey breadth (e.g., [61]). Additionally, stable isotope analysis [25,62] could also allow us to understand the long-term impacts of this predatory fish, considering that traditional dietary studies can only be seen as a snapshot of feeding behavior. Latitude was shown to influence the variation in *S. lucioperca* diet and likely reflects the foraging opportunism of the species and spatial changes in local prey supply. The most frequent and numerous preys found in the northern populations were *R. rutilus*, *S. lucioperca*, *M. salmoides* and Diptera, whereas in the south they were *A. alburnus*, *S. lucioperca*, Atyidae, *P. clarkii* and Diptera. Those spatial changes in the consumption of prey types are often indicative of changes in the abundance or availability of those prey items [36,59]. In fact, *R. rutilus* only occurs in northern populations of Douro, Ave and Cávado [30,63], while *A. alburnus* is a recent invader (last 20 years) to Portuguese watersheds having initially invaded southern rivers and expanded northwards [30]. Similarly, there was a high prevalence of pikeperch cannibalism (six populations), reaching up to 34% and 83% of preys in the lentic sites of Douro and Sado, respectively. This was already described in several studies (see [33–36,58,60]) and is related to local prey abundance. Moreover, lotic systems exhibit higher species diversity particularly in the lower reaches of the Vouga where the occurrence of migratory fish like Mugilids is prevalent. According to previous studies, see [64], a high species richness was described in the Tagus mainstem which is consistent with the higher prey diversity found in *S. lucioperca* in this site, observed as "Other Fish" (generally native fish). Finally, a similar pattern of increasing proportions of Atyidae in the diet has been observed in another top predator, the European catfish (*Silurus glanis*), in the lotic Tagus that exhibited lower fish prey richness and higher proportions of Crustaceans, see [65].

The influence of pikeperch size on its diet has often been reported [33,36], with a rapid replacement of macroinvertebrates by fish as the increasing size of pikeperch permits the handling of bigger, more profitable prey. While, in our results the pikeperch diet contained prey fish across the range of two size classes, the consumption of macroinvertebrates were most frequently encountered in the diets of smaller individuals. The absence of suitable-sized prey fish is likely to limit the pikeperch population's ability to reach piscivory, causing a delay to growth acceleration and a high mortality due to starvation and predation in size classes [62,66]. Cannibalism seems to occur throughout the life of pikeperch. Although its importance was found modest for individuals smaller than 25 cm SL, cannibalism became more important with increasing body size [36,60]. Moreover, we found evidence suggesting that both size classes of pikeperch in the Sado were cannibalistic. This might be explained by the time of sampling, as young-of-the-year (YOY) *S. lucioperca* would not have been present in the population due to the timing of spawning [67], but also because of the high density of juveniles in the population [68]. Nevertheless, we should not exclude limitations stemming from the stomach content analyses in providing accurate dietary assessments, as it was only completed at a single time of the year.

Pikeperch showed a reduction in population prey breadth which was accompanied by an increase in individual specialization. This indicates that the specialization observed at the population level was the result of individuals specializing on a subset of resources of the prey spectrum. Similar values for population diet breadth have been found in southern Europe, namely in the Alcántara reservoir in Spain [36]. The trends towards narrower diets and the use of a small array of prey by all individuals in Mediterranean rivers may reflect conditions of reduced intraspecific competition, thus facilitating the spread and integration of this invasive species [69]. Furthermore, the use of similar prey may also be associated with changes in fish assemblages across the latitude gradient, with a low diet breadth at population and individual levels favoring prey partitioning among species.

## 5. Conclusions

This work highlights that pikeperch is an opportunistic predator with a specialized feeding strategy that may potentially cause severe impacts on aquatic communities. Variation in diet was most strongly linked to latitude and ontogeny. Niche breadth remained narrow to minimize the diet variation among individuals and decrease the risk of competition for available resources. These findings are important for our efforts to maintain the integrity of the highly endangered native fish existing in Iberian watersheds, where native fish communities, originally devoid of any native predator, are highly vulnerable to new predators. In fact, it is urgent to evaluate the impact of these predators in riverine fish communities, generally dominated by natives, especially during their initial colonization (but see [65]). Describing the dietary traits of top predators, understanding their behavior and knowing their potential feeding preferences, are essential pieces of information to evaluate the predator impacts on our endemic fishes and enable a better conservation strategy focusing on these unique freshwater ecosystems.

**Author Contributions:** Conceptualization, D.R. and F.R.; methodology, D.R. and C.G.; investigation, D.R., C.G., J.G. and F.R.; writing—original draft preparation, D.R. and C.G.; writing—review and editing, J.G. and F.R.; visualization, C.G.; supervision, F.R.; funding acquisition, F.R. All authors have read and agreed to the published version of the manuscript.

**Funding:** This study was conducted in the frame of the projects SONICINVADERS (FCT ref. PTDC/CTA-AMB/28782/2017), FRISK (PTDC/AAG-MAA/0350/2014) and ISO-INVA (LISBOA-01-0145-FEDER-029105 + PTDC/CTA-AMB/29105/2017) co-funded by international funds through Lisboa 2020—Programa Operacional Regional de Lisboa, in its FED ER component (Project ref. LISBOA-01-0145-FEDER-029105) and national funds through FCT—Fundação para a Ciência e a Tecnologia, I.P (Project ref. PTDC/CTA-AMB/29105/2017). Additional support was provided by FCT through the strategic plan of the Marine and Environmental Sciences Centre (MARE) (UID/Multi/04326/2019).

**Institutional Review Board Statement:** Not applicable.

**Informed Consent Statement:** Not applicable.

**Data Availability Statement:** The data presented in this study are available on request from the corresponding author. The data are not publicly available due to privacy restrictions.

**Acknowledgments:** The authors would like to thank all the fishermen that contributed with samples that made it possible to perform this study, in particular to Carlos Serras, João Lobo and Francisco Pinto. A special thanks also, to Luís da Costa for the graphic development of Figure 1 and to the anonymous reviewers for their helpful contribution. This work was supported by project "SONICINVADERS", co-funded by Portugal 2020 and Alentejo2020 through EU ERDF funds (ALT20-03-0145-FEDER-028782), and by national funds through FCT (PTDC/CTA-AMB/28782/2017). Additional support was provided by FCT through the strategic plan of the Marine and Environmental Sciences Centre (MARE) (UID/MAR/04292/2013).

**Conflicts of Interest:** The authors declare no conflict of interest.

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
