# Peer review of "Variation in Diet Patterns of the Invasive Top Predator Sander lucioperca (Linnaeus, 1758) across Portuguese Basins"

_water, doi:10.3390/w13152053_

Round 1

Reviewer 1 Report

This work examines the diet of an invasive fish species in freshwater habitats. Although it has been carefully performed, it is based on very small sample size, the definition of two size classes is not explained at all,  and the examined fish species has been previously studied; accordingly, little novelty is added by the MS that could interest readers at wide-scale. 

Based on above I am suggesting a rejection of this MS. Authors could possible re-submit their work to another, more relevant, (i.e locally focused) journal.  

Reviewer 2 Report

The aim of this manuscript is to compare the diet of pikeperch across different populations in Portugal using a spatial approach. The manuscript is well writing, I have just few specific comments on it.

Below I address my specific comments point-by-point. I hope you find these comments helpful in revising your manuscript.

Specific comments:

First there are lot of double space, please correct them : l 57, l 60, l 75, l 105, l 215, l 279, l 288, l 423

Please check in all text the p-value format : P or p, in italic or not.

L 71: Sampling were realized between April and October, have you consider the time when you analyzed your data ? Is it possible that stomach contents were different in function of the time period and can explain the presence or absence of certain prey in pikeperch stomach ? And, what about the period from November to March ? Maybe the prey diversity is underestimate ?

L 103: please add quotation marks for other insects as “other fish”

L 142: it misses formula/ equation

Table 2 and associated text (l176 to 179): when we consider an interaction between two factors, and when this interaction was significant, it is not possible to consider simple effect. Consequently, in your table and in your text, please presented only the interaction or, at least, the interaction first and then the simple effects (but you can’t discuss simple effects with a significant interaction).

Figure 3. I don’t understood why Rivers are in color legends while they were already represented on abscissa axis ?

L252: It is strange to begin a sentence with article number. Is it possible to rewrite this sentence please ?

Reviewer 3 Report

The article contains new and interesting data on the diet of pikeperch in different types freshwater bodies of Portugal. The pikeperch is expanding its distribution in Portugal, and being a top predator, it can have a great impact on the populations of native fish species and other animals. Therefore, information about feeding is important for assessing the possible effects that pikeperch can cause to native species. The information provided in the article is also important for understanding the general patterns of invasion of predatory fish into new areas. The article is well written. The main comments of reviewer are related to the chapter "Results".

  1. Data on the diet of pikeperch of different sizes are not provided. It is only briefly mentioned that, in general, there is a shift from feeding mainly on invertebrates to feeding on fish. But this is not enough. It is necessary to show what exactly fish of different sizes ate in different reservoirs and how obvious these shifts were. It is also necessary to say what age the fish studied were (pikeperch).
  2. The article casually mentions that the pikeperch consumed conspecifics, among other preys. Cannibalism is really characteristic of pikeperch. This is an interesting phenomenon. The authors should present the results on cannibalism – at what size of pikeperch it was observed, what proportion could consist of conspecifics in diet, and more.
  3. The authors talk about certain patterns of diet changes depending on the geographical latitude of the waterbodies and its type (lotic\lentic), but the data on this issue are presented in a too generalized form (in the form of Figure 2 or tables with statistical results – Tables 2 and 3).
  4. Table 1 presents the results concerning the diet in different waterbodies of all the studied pikeperches, the length of which varied in a wide range – from 9.4 cm to 60.3 cm. It is clear that the nutrition of a small pikeperch (up to 15-20 cm) was completely different than that of a large one (more than 40 cm). But what size were the studied pikeperch in different waterbodies? Such a table gives only the most general ideas about the diet of pikeperch.
  5. There are a large number of invasive fish in the freshwaters of Portugal. It would be interesting if authors analyzed their data in such a way as to show what proportion of invasive and native species make up in the diet of pikeperch.
  6. The authors write that " ... Prior to analysis, animal prey proportions were determined for each individual in relation to the total prey in its stomach...." (lines 114-115). This is difficult to understand, since all the preys of the pikeperch were animals.
